# Molecular Detection of Metronidazole and Tetracycline Resistance Genes in *Helicobacter pylori*-Like Positive Gastric Samples from Pigs

**DOI:** 10.3390/antibiotics12050906

**Published:** 2023-05-13

**Authors:** Francisco Cortez Nunes, Emily Taillieu, Teresa Letra Mateus, Sílvia Teixeira, Freddy Haesebrouck, Irina Amorim

**Affiliations:** 1School of Medicine and Biomedical Sciences (ICBAS), University of Porto, 4050-313 Porto, Portugal; up201711052@up.pt (F.C.N.); up201006078@up.pt (S.T.); iamorim@ipatimup.pt (I.A.); 2Institute for Research and Innovation in Health (i3S), University of Porto, 4200-135 Porto, Portugal; 3Institute of Molecular Pathology and Immunology of the University of Porto (IPATIMUP), 4200-135 Porto, Portugal; 4Department of Pathobiology, Pharmacology and Zoological Medicine, Faculty of Veterinary Medicine, Ghent University, B9820 Merelbeke, Belgium; freddy.haesebrouck@ugent.be; 5CISAS-Centre for Research and Development in Agrifood Systems and Sustainability, Escola Superior Agrária, Instituto Politécnico de Viana do Castelo, 4900-347 Viana do Castelo, Portugal; tlmateus@esa.ipvc.pt; 6EpiUnit—Instituto de Saúde Pública da Universidade do Porto, Laboratory for Integrative and Translational Research in Population Health (ITR), Rua das Taipas, n° 135, 4050-091 Porto, Portugal; 7Veterinary and Animal Research Centre (CECAV), UTAD, Associate Laboratory for Animal and Veterinary Sciences (AL4AnimalS) Quinta de Prados, 5000-801 Vila Real, Portugal

**Keywords:** antimicrobial resistance, antimicrobial resistance gene, gastric helicobacters, PCR, Sus scrofa, One Health

## Abstract

Antimicrobial resistance is a major public health concern. The aim of this study was to assess the presence of antibiotic resistance genes, previously reported in *Helicobacter pylori,* in gastric samples of 36 pigs, in which DNA of *H. pylori*-like organisms had been detected. Based on PCR and sequencing analysis, two samples were positive for the *16S rRNA* mutation gene, conferring tetracycline resistance, and one sample was positive for the *frxA* gene with a single nucleotide polymorphism, conferring metronidazole resistance. All three amplicons showed the highest homology with *H. pylori*-associated antibiotic resistance gene sequences. These findings indicate that acquired antimicrobial resistance may occur in *H. pylori*-like organisms associated with pigs.

## 1. Introduction

Over 50% of the world population is infected with the Gram-negative bacterium *Helicobacter pylori* (*H. pylori*), one of the main causes of acute and chronic gastritis, peptic ulcer disease, gastric adenocarcinoma and gastric mucosa-associated lymphoid tissue (MALT) lymphoma, which can also cause extra-gastrointestinal diseases [1,2,3]. Humans can also be infected with gastric non-*Helicobacter pylori Helicobacter* (NHPH) species, with an estimated global prevalence of 0.2–6% in patients undergoing a gastroscopy [4,5] and a prevalence of 20.8–29.1% in selected *H. pylori*-negative gastric patient cohorts [6,7]. Besides a pathophysiological involvement in gastric disease, gastric NHPHs have also been associated with extra-digestive diseases [2].

In order to manifest gastric disease, these *Helicobacter* species possess several virulence factors involved in colonizing the gastric niche (e.g., urease), inducing pathology and evading the immune system to promote persistence of infection. Although two major cytotoxic virulence factors in *H. pylori*, cytotoxin-associated gene pathogenicity island (cagPAI) and vacuolating cytotoxin A (VacA), appear to be absent in gastric NHPHs, gamma-glutamyltranspeptidase (GGT) is suggested to play an important pathophysiological role [8].

In animals, there are several studies reporting infections with helicobacters and their association with gastric alterations [4,9,10,11,12]. Among these, there are sporadic reports of infections with *H. pylori*-like organisms in pigs [13,14,15,16,17,18]. These *H. pylori*-like organisms appear to carry genes similar to *H. pylori*, including the *ureAB* gene. However, no amplification of the *glmM* (*ureC*) gene was achieved in these pig samples positive for a *H. pylori*-specific, *ureAB*-based PCR assay. Therefore, the term *H. pylori*-like organisms is used [18].

In humans, there are different therapeutic approaches to *H. pylori* infections, although standard treatment consists of triple therapy that includes a proton-pump inhibitor and two antibiotics. Among the antibiotic choices, amoxicillin, clarithromycin and metronidazole are the most commonly used, being considered the first line of treatment [1,19,20,21,22,23,24].

Antimicrobial resistance (AMR) is one of the most concerning One Health issues worldwide [25]. The World Health Organization (WHO) has classified the most important resistant bacteria at a global level for which there is an urgent need for new treatments [26]. This classification was done according to the species and the type of resistance resulting in three priority tiers: critical, high and medium, where *H. pylori* is classified as high risk, specifically for clarithromycin-resistance [26]. In the most recent management guidelines for *H. pylori* infection (Maastricht V/Florence Consensus Report), the increasing resistance to existing antibiotic regimens was one of the major concerns raised [27].

Based on the importance of antimicrobials for treating human infections and the antibiotics used in veterinary medicine, the WHO published a priority list of antimicrobials grouped into three categories: (1) Critically important, subdivided into highest priority and high priority, (2) Highly important and (3) Important [28]. Similarly, the World Organization for Animal Health (WOAH) developed a list of antimicrobials of veterinary importance further classified as: veterinary critically important, veterinary highly important, and veterinary important antimicrobials [29]. Within these lists, amoxicillin is classified as critically important (WHO, WOAH), clarithromycin as critically important (WHO), tetracycline as highly important (WHO, WOAH), and metronidazole as important (WHO) [28,29].

Penicillins and tetracyclines are among the most commonly used antimicrobial classes in pigs [30,31]. The use of antimicrobials has been associated with direct and indirect impacts on the gastrointestinal microbiota and its antimicrobial resistome [30]. This has raised public health concerns due to selective pressure on opportunistic pathogens.

The main mechanism that contributes to *Helicobacter* resistance development is the acquisition of point mutations in the DNA. In the specific case of *H. pylori*, it acquires resistance via chromosomal mutations and horizontal transfer of resistance genes [32,33].

Regarding metronidazole, a nitroimidazole that acts as a bactericidal agent by interacting with a nitroreductase homolog, *rdxA*, resistance to it in *H. pylori* has been linked to mutations in the gene *rdxA*, while changes in the gene *frxA*, which encodes for NADH:flavin oxidoreductase, have also been implicated [32].

Tetracyclines bind to the ribosomal 30S subunit, inhibiting protein synthesis. Tetracycline resistance can be acquired in the majority of bacterial species through efflux systems or through ribosomal protection proteins. A reduction in membrane permeability, changes in ribosome binding, enzymatic antibiotic degradation, active efflux and changes in membrane permeability all appear to play a role in tetracycline resistance. In *H. pylori*, resistance to tetracyclines seems to be conferred by mutations in the *16S rRNA* gene [34].

The aim of this study was to assess the presence of gene mutations associated with resistance to amoxicillin, metronidazole, clarithromycin and tetracycline in porcine gastric samples that were shown to be positive for *H. pylori*-like DNA in a previous study [18].

## 2. Results

### 2.1. PCR Results

Out of the 36 *H. pylori*-like positive tested samples, three *pars oesophagea* samples (8.3%) were PCR-positive for genes conferring resistance to antimicrobials. Two were found PCR-positive for the *16S rRNA* mutation gene conferring tetracycline resistance and one was found PCR-positive for the *frxA* gene that can be associated with metronidazole resistance in the presence of a single nucleotide polymorphism (SNP) (Table 1; See Appendix A for gel electrophoresis photos).

### 2.2. Sequencing and Sequence Analysis of Positive PCR Products

The bidirectional sequencing and basic local alignment search tool (BLAST) analysis of consensus sequences of partial *16S rRNA* mutation gene amplicons showed an identity ranging from 98.6–100% with *H. pylori* (accession nr. OP389222) *16S rRNA* mutation conferring resistance to tetracycline (ARO:3003510) for both positive samples.

The other positive sample was also subject to bidirectional sequencing and BLAST analysis of the consensus sequence. The amplicon showed an identity of 99.85% to *H. pylori* (accession nr. CP026515). The obtained sequence was also analyzed using the Resistance Gene Identifier (RGI) to predict resistomes from nucleotides on homology and SNP models. The RGI criteria fully corresponded with a *H. pylori frxA* mutation conferring resistance to metronidazole, with SNP Y62D, with an identity of the matching region of 99.07% (ARO:3007059) [35,36,37] (See Appendix A for more sequence analysis details).

To support these results, phylogenetic trees were constructed using the Neighbor-Joining method. The bootstrap consensus trees can be found in the Appendix A.

## 3. Discussion

Our results showed that AMR-associated mutation genes presenting the highest homology with *H. pylori*-associated genes occurred in three samples from the stomach of pigs, that were previously reported as *H. pylori*-like positive.

To achieve these results, we partly relied on the methodology described by Diab et al. (2018) [38] and Lee et al. (2018) [39] who also aimed to detect antibiotic resistance genes (ARGs) of *H. pylori* conferring resistance to clarithromycin, metronidazole, amoxicillin and tetracycline, however, in human patients. The former examined gastric biopsy specimens from patients found positive based on rapid urease testing and the presence of *H. pylori 16S rRNA*. The latter research was performed using *H. pylori* isolates obtained from patients. AMR gene PCR positive amplicons were in both studies also sequenced and subjected to BLAST for sequence and mutation analysis, but not analyzed using the Comprehensive antibiotic resistance database (CARD). Although AMR in *H. pylori*-like positive samples had not been studied yet, there are other studies that investigated the same genes in other animal samples positive for other *Helicobacter* species [40,41,42].

In our study, two of the analyzed porcine gastric tissue samples from the *pars oesophagea* were positive for the *16S rRNA* mutation gene conferring resistance to tetracycline. Another sample of the *pars oesophagea* was positive for the *frxA* gene with SNP Y62D mutation conferring resistance to metronidazole. The three samples were subject to BLAST with homologies with *H. pylori* ranging from 98.6 to 100%. In humans, metronidazole is regarded as one of the drugs of choice in triple therapy against *H. pylori* infections. The overall prevalence of *H. pylori* resistance to metronidazole was found to be 47.2%, which is highest in Africa (75.0%), followed by South America (52.9%), Asia (46.6%), Europe (31.2%) and North America (30.5%) [43], implicating reduced treatment efficacy in humans [20]. In addition, tetracyclines are also commonly used in rescue eradication regimens against *H. pylori*, displaying a worldwide resistance rate of 11.7% [43]. Of note, none of the included samples were positive for *H. pylori 23S rRNA* potentially associated with clarithromycin conferring point mutations. Since this is a *H. pylori* species-specific PCR assay (not cross-reacting with *23S rRNA* gene sequences of other gastric NHPHs according to in silico analysis), this may further confirm our previous findings regarding the presence of *H. pylori*-like DNA in porcine gastric samples rather than DNA of *H. pylori* itself.

Although the use of antibiotics for growth promotion is prohibited in several countries, including European Union member states, tetracyclines are still used as a growth promoter in many countries. In pigs, tetracyclines are generally the most commonly used antibiotics [31,42,44,45]. Tetracycline resistance genes are some of the most abundant ARGs in the pig microbiome [31,46]. A study conducted by Liu et al. (2019) detected tetracycline ARGs in all evaluated (22/22) pork samples [42]. Ricker et al. (2020) isolated and extracted DNA from porcine feces for retrospective ARG analysis. They reported that the use of tetracyclines in pigs promotes co-selection for resistance genes for aminoglycosides and tetracyclines [47], although this has not been described in *Helicobacter* species.

Apart from our study, acquired resistance to tetracyclines has also been reported to occasionally occur in *H. suis* isolates obtained from the stomach of pigs [40].

During the processing of pig carcasses, the surface of the carcass and the parts destined for retail sale can get contaminated with organisms coming from the hide/skin of the animals, gut content, workers’ hands, and the slaughter environment [48]. This can predispose humans to contact with AMR pathogens and ARGs as stated by Liu et al. (2019). This suggests that ARGs, could be potentially transmitted to humans via the meat industry chain/feed supply, pig feeding and pork production [42].

Furthermore, antibiotic use eliminates susceptible pathogens allowing resistant strains to continue to evolve and multiply. Selective pressure from antimicrobial exposure thus provides resistant pathogens with an evolutionary advantage and favors their spread [30].

Although our results point towards the presence of ARGs in *H. pylori*-like organisms conferring resistance to tetracycline and metronidazole, the interpretation of these findings should be done with caution. The significance for human and animal health is not yet completely clear, since there are no reports of *H. pylori*-like infections in humans and the relevance of *H. pylori*-like organisms in both animals and humans requires further investigation. Studies should be conducted with a larger sample size and ideally, isolation of the *H. pylori*-like bacteria should be performed in order to characterize the organisms and test for antimicrobial susceptibility in depth.

## 4. Materials and Methods

### 4.1. Sample Selection

DNA extracts from 36 gastric samples from pigs (29 samples of the *pars oesophagea* and 7 samples of oxyntic mucosa), containing *H. pylori*-like DNA as shown by PCR and sequencing analysis (see Appendix A for these PCR details), were analyzed for the presence of *H. pylori*-specific ARGs [16,17,18].

### 4.2. PCR Conditions and Sequencing

The *H. pylori*-like positive samples were subjected to conventional PCR assays to test for the presence of genes related to AMR in *H. pylori*, including *Pbp1A* (amoxicillin), *rdxA* and *frxA* (metronidazole), *16S rRNA* mutation gene (tetracycline) and *23S rRNA* (clarithromycin) in order to identify point mutations (Table 2).

Aliquots of each PCR product were electrophoresed on 1.5% agarose gel stained with Xpert Green Safe DNA gel stain (GRISP, Porto, Portugal) and examined for the presence of specific fragments under UV light. DNA fragment size was compared with the standard molecular weight, 100bp DNA ladder (GRISP, Porto, Portugal). For the negative control, distilled water was used. No positive control was used on PCR to test resistance genes.

The amplicons of each positive sample were sequenced. Bidirectional sequencing was performed using the Sanger method at the Genomics core facility of the Institute of Molecular Pathology and Immunology of the University of Porto, Portugal. Sequence editing and multiple alignments were performed with the MegaX Molecular Evolutionary Genetic Analysis version 10.1.8 [49]. The sequences obtained were subject to BLAST analysis using the non-redundant nucleotide database (http://blast.ncbi.nlm.nih.gov/Blast.cgi (accessed on 22 September 2022)) [36,37]. Sequences were also analyzed through the CARD to identify additional antibiotic-resistant gene mutations [35].

## 5. Conclusions

AMR is a One Health concern mainly related to the animal and human use of antibiotics. Our results demonstrate that AMR may occur in *H. pylori*-like organisms from pigs since we identified *H. pylori*-associated *16S rRNA* mutation genes conferring tetracycline resistance and a mutation in the *frxA* gene that may confer metronidazole resistance in three different *pars oesophagea* samples included in the current study. The significance of these findings for public and animal health requires further investigation, including attempts to isolate and in-depth characterize these organisms and to determine their possible pathogenic significance.

## Figures and Tables

**Table 1 antibiotics-12-00906-t001:** PCR results regarding AMR genes per gastric region.

*H. pylori*-like Positive Samples	*frxA* Gene PCR Positive (n/N) (%)	*rdxA* Gene PCR Positive (n/N) (%)	*16S rRNA* Mutation Gene PCR Positive (n/N) (%)	*23S rRNA* Gene PCR Positive (n/N) (%)	*Pbp1A* Gene PCR Positive (n/N) (%)
*Pars oesophagea* (N = 29)	1/29 (3.4%)	0/29 (0.0%)	2/29 (6.9%)	0/29 (0.0%)	0/29 (0.0%)
Oxyntic mucosa (N = 7)	0/7 (0.0%)	0/7 (0.0%)	0/7 (0.0%)	0/7 (0.0%)	0/7 (0.0%)

**Table 2 antibiotics-12-00906-t002:** Primer sequences and thermocycling conditions for detection of genes and mutation genes conferring resistance to antimicrobials.

Antimicrobials		Sequence	Target Gene	Thermo Cycle Conditions			Reference
				Temp. (°C)	Time	Nr. Cycles	
Amoxicillin	Forward	GCG ACA ATA AGA GTG GCA	*Pbp1A*	95 95 56 72 72	3′ 1′ 1′ 5′ 10′	35	[38,39]
	Reverse	TGC GAA CAC CCT TTT AAA T					
Metronidazole	Forward	AAT TTG AGC ATG GGG CAG A	*rdxA*	95 94 60 72 72	5′ 30" 30" 1′ 10′	35	[38,39]
	Reverse	GAA ACG CTT GAA AAC ACC CCT					
	Forward	TGG ATA TGG CAG CCG TTT A	*frxA*	95 95 58 72 72	5′ 30" 30" 1′ 10′	35	[38,39]
	Reverse	GGT TAT CAA AAA GCT AAC AGC G					
Tetracycline	Forward	CGG TCG CAA GAT TAA AAC	*16S rRNA* *mutation*	95 95 55 72 72	10′ 5" 2" 30" 10′	45	[38]
	Reverse	GCG GAT TCT CTC AAT GTC					
Clarithromycin	Forward	TCA GTG AAA TTG TAG TGG AGG TGA AAA	*23S rRNA*	95 92 60 72 72	10′ 15" 1′ 1′ 10′	40	[38]
	Reverse	CAG TGC TAA GTT GTA GTA AAG GTC CA					

## Data Availability

The data presented in this study are available on request from the corresponding author.

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
