# Peer review of "Molecular Detection of Metronidazole and Tetracycline Resistance Genes in Helicobacter pylori-Like Positive Gastric Samples from Pigs"

_antibiotics, 2023, doi:10.3390/antibiotics12050906_

Round 1
Reviewer 1 Report
The paper submitted for review relates to the medically important issue of antibiotic resistance.
In the introduction, the authors clearly presented the issues associated with the subject of the publication
The results of the work are presented in Table 1.
In the paper, the authors cited 49 items of current scientific bibliography.
On the basis of the obtained research results, they drew appropriate and logical conclusions.
Comments
In my opinion, in the classical layout of scientific papers, the methods should be in section 2 after the introduction.
In my opinion, the work can be accepted for publication after minor (suggested additions).
Author Response
The authors sincerely thank the reviewer for the very kind comments and suggestions. Accordingly, please find our response below:
We acknowledge that it is atypical to provide the methods between the discussion and the conclusion section. However, for this manuscript a template suggested by the journal was used which specifically asked to report it in this manner.
Reviewer 2 Report
1. The introduction should be complemented, including the pathophysiology of H. pylori disease.
2. Change references before 2003 for recent references.
3. Add PCR and sequencing results as supplementary material.
4. Rewrite the conclusion highlighting the main findings.
Revise grammatical language aspects.
Author Response
The authors sincerely thank the reviewer for the very kind comments and suggestions. Accordingly, please find our response below:
1. The introduction has been supplemented with additional relevant background information regarding H. pylori and the pathophysiology of gastric Helicobacter species. The corresponding modifications were made in red (track changes p. 2 lines 46-52 and 70-73).
2. References 36 and 37 indeed date from 1990 and 2002, respectively. However, these are original references to the NCBI BLAST tool and GenBank to support the methods applied in order to retrieve the corresponding results.
3. The PCR and sequencing results have been added in a Supplementary Materials file (Figures S1. – S9.) and are referred to in-text (p. 3 lines 113-114 and line 128).
4. The conclusion has been rewritten in part in order to include our main findings. The corresponding modifications were made in red (track changes p. 6 lines 224-226).
Reviewer 3 Report
Dear Authors,
Please explain in the material and methods the data information of the 36 gastric samples from pig used in this study and what is the primer used to confirm that you got helicobacter plyori .
Please provide us by phot of PCR results of the resistance genes .
In the resistance genes results you have to create a phylogentic tree.
Author Response
The authors sincerely thank the reviewer for the very kind comments and suggestions. Accordingly, please find our response below:
- The PCR details for the detection of H. pylori-(like organisms) used in our previous study as mentioned in the current manuscript have been added in a Supplementary Materials file (Table S1.) and are referred to in-text (p. 5 line 197).
- Photos of the PCR results and screenshots of the sequencing analyses have been included in a Supplementary file (Figures S1. – S9.) and are referred to in-text (p. 3 lines 113-114 and line 128).
- Phylogenetic trees for the identified antibiotic resistance genes have been included in a Supplementary Materials file (Figures S10. and S11.) and are referred to in-text (p. 3 lines 129-131).